# Numerical Simulation of the Hydraulic Characteristics and Fish Habitat of a Natural Continuous Meandering River

**Pingyi Wang** [1,2,3], **Jian Li** [1,2,3,*] 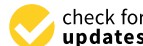, **Meili Wang** [1,2], **Jielong Hu** [4] **and Fan Zhang** [1,2,3]

1    Key Laboratory of Ministry of Education of Water Resources and Water Transportation Engineering, Chongqing Jiaotong University, Chongqing 400074, China
2    National Engineering Technology Research Center for Inland Waterway Regulation of Chongqing, Chongqing Jiaotong University, Chongqing 400074, China
3    College of Hehai, Chongqing Jiaotong University, Chongqing 400074, China
4    Tianjin Research Institute for Water Transport Engineering, Tianjin 300456, China
*    Correspondence: linsanity0920@163.com

**Highlights:**

- To study the hydraulic characteristics of natural continuous curved river flows;
- To study the influence of the hydraulic characteristics of natural continuous curved river flows on fish habitats;
- To evaluate the impact of channel regulation on the construction of ecological waterways.

**Abstract:** Increasing international attention is being focused on the construction of ecological waterways. Ecological waterways are not only able to successfully sustain the navigational functions for ships, but also provide a stable long-term environment in which fish can survive. In nature, meandering rivers are the most common form. Compared with straight rivers, their flow conditions are complex and not conducive to ship navigation and fish survival. There has been less research conducted on the construction of ecological waterways in naturally curving rivers. Therefore, in this paper, a naturally continuous curved river was used as a research object, and numerical simulation was employed to study the hydraulic characteristics of the river and the survival environment by fish by introducing the Shannon diversity index. It was concluded that a naturally continuous curved river cannot meet the navigational requirements and does not provide a stable survival environment for fish. After proposing a targeted channel restoration plan, the hydraulic characteristics of the flow were significantly improved, and the flow velocity Shannon diversity index was reduced. The restored channel met the navigational requirements of vessels, and provided a more stable environment for fish. In the construction of the continuous bend river ecological corridor, the location of the continuous bend stream connection was used as a key restoration area. This paper provides ideas and basic research for the sustainable development of ecological river channels and the construction of continuous curved river channels.

**Keywords:** continuous meandering streams; numerical simulation; hydraulic characteristics; ecological channel regulation; ecological sustainability; Shannon diversity index

## 1. Introduction

The Earth is rich in water resources. The different natural geographical locations, landforms, and climatic conditions lead to different characteristics of river morphology. Among them, curved rivers are the most common river type. The Lumber River in the United States, the Klarälven River in Sweden, and the Jingjiang River in China are examples of curved rivers [1]. Researchers have studied bend flow structures through experiments or numerical simulations. It was found that among the main flows along the flow direction, the lateral and vertical secondary flows caused by bends are very important in analyzing

turbulence structure [2]. There is a strong correlation between the flow characteristics and geometry in bends [1]. The hydrodynamic instability of the curved channel is related to the degree of bend, incoming flow, and the width/depth ratio. Under different aspect ratios, the plane flow field and bed stress distribution of bend flows are different [3]. Through numerical simulation (LES and RANS), researchers have compared and studied the main flow, secondary flow, and turbulence in curved open channel flow with different water depths [4,5]. The hydrodynamic characteristics of curved rivers sustaining plants can be analyzed through physical model tests. It has been found that the existence of evenly distributed vegetation on the concave bank will weaken the secondary flow [6].

In ecological channel construction, improving the unstable flow characteristics of the curved channel is very important. In particular, constructing ecological channels near the sea entrance significantly impacts navigation and transportation, flood discharge, and coastal port construction [7]. Traditional waterway regulation methods have recently been abandoned, and higher ecological waterway demand has been put forward. The definition of an ecological waterway is to create an ideal river ecosystem with natural stability, health, openness, and harmony between people and water through various engineering and technical means, such as waterway regulation to ensure safe navigation. As key aquatic animals, fish require a stable living environment, which is an important factor of ecological waterway construction. Fish inhabit specific water flow conditions. In order to study the response mechanisms of fish to water flow, researchers have taken the hydraulic characteristics of water flow as the corresponding parameters of fish to classify the swimming speed and swimming state of fish, in which the critical speed and explosive swimming speed are the main research objects [8–15]. To some extent, fish movement behaviors directly reflect the advantages and disadvantages of the ecological river channel. Yang et al. [16] established a two-dimensional numerical flow model from Nanjing to the estuary, and studied the overall effect of the secondary phase of the remediation project, providing technical support for the construction of a sustainable ecological waterway. Pingyi Wang and Li Jian et al. [17] aimed to provide a stable living environment for aquatic animals and plants; they studied the hydraulic flow characteristics of ecological slope protection structures through physical model tests. Kuhnle [18] et al. showed that ding–dam complexes are common riparian protection structures and can be used in the construction of ecological waterways for aquatic organisms because they create and maintain good habitat conditions. Kang [19] and others found that the dammed fields created by the establishment of dingbats could provide a suitable habitat for fish by model tests.

In summary, at this stage, researchers have conducted a large number of theoretical studies on ecological rivers; although, these studies have mainly focused on smooth and straight rivers, and less on natural rivers, especially curved rivers. In nature, there are many obstructive factors in natural rivers, which restrict the safe navigation of ships and cannot provide a good living environment for aquatic plants and animals, which seriously affects the construction and sustainable development of ecological waterways. Here, the authors selected a representative natural continuous curved river channel as the research object, and used numerical simulation as the study method (using the numerical simulation software, Mike21, 2014, DHI, Denmark) to introduce the Shannon diversity index to analyze whether the hydraulic characteristics in natural continuous curved river channels meet the requirements for the safe navigation of ships and provide a stable survival environment for fish. The authors propose a reasonable adjustment plan to address the disadvantages factors that are not conducive to the construction and sustainable development of ecological river channels, and compare the hydraulic characteristics of the river before and after remediation to analyze whether the requirements for safe navigation of ships and long-term survival of fish are met. At the same time, in the construction of the eco-channel of the continuous curved river, the key remediation areas for the construction of the eco-channel were determined according to the hydraulic characteristics. The main objective of this study was to provide basic research and new ideas for the direction of the sustainable

development of ecological waterways in natural rivers through study of the construction and sustainable development of ecological waterways in natural rivers.

## 2. Materials and Methods

### 2.1. Mathematical Model

2.1.1. Hydrodynamic Control Equations

The mathematical model was established by relying on the Navier–Stokes equations with Reynolds value distribution; it was incompressible in three terms and obeyed the Boussinesq assumption and the assumption of hydrostatic pressure. For the continuous curved channel in this study, the horizontal direction of its motion scale was much larger than the vertical direction; thus, in the numerical simulation calculation, the extended water depth direction was the vertical direction, the average value of which was selected for calculation, although in the vertical direction of its dynamic water pressure to meet the law of hydrostatic pressure.

The continuity equation is:

$$\frac{\partial h}{\partial t} + \frac{\partial h\overline{u}}{\partial x} + \frac{\partial h\overline{v}}{\partial x} = hS \tag{1}$$

The equation of motion in the $x$ and $y$ directions is:

$$\frac{\partial h\overline{u}}{\partial t} + \frac{\partial h\overline{u}^2}{\partial x} + \frac{\partial h\overline{v}\overline{u}}{\partial y} = f\overline{v}h - gh\frac{\partial \eta}{\partial x} - \frac{h}{\rho_0}\frac{\partial p_\alpha}{\partial x} - \frac{gh^2}{2\rho_0}\frac{\partial p}{\partial x} + \frac{\tau_{sx}}{\rho_0} - \frac{\tau_{bx}}{\rho_0} - \frac{1}{\rho_0}\left(\frac{\partial s_{xx}}{\partial x} + \frac{\partial s_{yy}}{\partial x}\right) + \frac{\partial}{\partial x}(hT_{xx}) + \\ \frac{\partial}{\partial y}(hT_{yy}) + hu_sS \tag{2}$$

$$\frac{\partial h\overline{v}}{\partial t} + \frac{\partial h\overline{v}^2}{\partial x} + \frac{\partial h\overline{v}\overline{u}}{\partial y} \\ = f\overline{v}h - gh\frac{\partial \eta}{\partial y} - \frac{h}{\rho_0}\frac{\partial p_\alpha}{\partial y} - \frac{gh^2}{2\rho_0}\frac{\partial p}{\partial y} + \frac{\tau_{sy}}{\rho_0} - \frac{\tau_{by}}{\rho_0} - \frac{1}{\rho_0}\left(\frac{\partial s_{yx}}{\partial x} + \frac{\partial s_{yy}}{\partial x}\right) \\ + \frac{\partial}{\partial x}(hT_{xy}) + \frac{\partial}{\partial y}(hT_{yy}) + hv_sS \tag{3}$$

In the above formula, $t$ represents time; $\eta$ represents the water level; $h$ represents the total water depth; $u$ and $v$ represent velocity components in $x$ and $y$ directions, respectively; $f$ represents the Coriolis force; $S$ stands for source item; $g$ represents the gravitational acceleration; $\rho$ represents the water density; $\overline{u}$ and $\overline{v}$ represent the average velocity along the vertical direction; $T_{ij}$ is the transverse stress; and $\tau$ is the riverbed stress.

2.1.2. Boundary Conditions and Model Parameter Settings

During the numerical simulation calculations, the authors set the inlet to have a specified discharge value, the outlet to a specified water level, and both banks to zero-velocity land. To avoid instability in the model calculations, the dry water depth, flooding depth, and wetting depth were set as 0.005 m, 0.05 m, and 0.1 m, respectively. Additionally, the eddy viscosity coefficient was considered to be a constant 0.28. The resistance coefficient varied depending on the location, and was assigned values from 0.01 to 0.05 [20,21].

2.1.3. Grid Division

In this study, the topographic data of the natural river channel were acquired by an unmanned survey vessel equipped with a sounding system (multibeam system). The underwater riverbed topography data were initially surveyed and calibrated to a high degree of accuracy. The riverbed topography was mapped using professional CAD drafting software. After the authors had determined the extent of the numerical simulation of the natural continuous curve channel from the riverbed topography map, the riverbed boundary and topographic elevation data were imported into MIKE21 grid-mapping software.

Considering the characteristics of the continuous bend, the authors used the unstructured grid finite volume method to discretize and solve the 2D shallow water equations in

this numerical simulation [22–24]. About 31,166 grids and 16,150 grid nodes were arranged in this numerical simulation.

## 2.2. The Shannon Diversity Index

The Shannon diversity index was introduced in this study. The authors determined the Shannon diversity index for river flow velocity based on the distribution of the river flow velocity. By looking at the flow Shannon diversity index, the authors analyzed whether ship navigation and fish habitats had been improved.

Shannon [25] introduced the concept of entropy into information theory to represent the average amount of multiple information sources sent by a certain signal source. Information entropy is a measure of uncertainty of discrete random variables. The calculation formula is as follows:

$$H(X) = E(I(x_i)) = \sum_{i=1}^{n} p(x_i) \log \frac{1}{p(x_i)} = -\sum_{i=1}^{n} p(x_i) \log(x_i) \tag{4}$$

where $p(x_i)$ is the event $x_i$ in the probability system. $I(x_i)$ is the probability of occurrence. The amount of information contained in the occurrence of the event is represented by $I(x_i) = \log(1/p((x_i))$. In practical calculations, natural logarithms based on a constant, e, are often taken. The more uncertain and complex the random events are, the greater the information entropy is, and vice versa.

This information entropy was later called the Shannon diversity index. The Shannon diversity index can express system complexity and diversity; therefore, it is widely used in ecology, landscape ecology, and other fields to reflect biodiversity and heterogeneity. In this study, the Shannon diversity index was selected to take the hydraulic characteristics of continuous meandering channel flow as the index to measure the degree of patch differentiation, and the channel velocity diversity index, $H_v$, was obtained according to the calculation results of the index:

$$H_v = -\sum_{i=1}^{n} P_{vi} ln(P_{vi}) \tag{5}$$

where $H_v$ is the velocity diversity index and $P_{vi}$ is the ratio of the patch area of the $i$ velocity to the total area of the calculation area.

The Shannon diversity index was also introduced to add a new reference factor for the sustainable development of ecological waterways.

## 2.3. Experimental Parameter Selection

The river channel in this study was a continuous curved river channel, in its original undeveloped state. The river channel trend was "m" type, and the bank slopes and riverbed landforms on both banks were greatly affected by the water flow. Navigation obstruction points/factors and uneven riverbed terrain lead to extremely complex flow hydraulic conditions. The total length of the river was about 2.7 km, composed of curve I and curve II. Here, the length of curve I was about 1.3 km, and there were navigation obstacles and low depressions on both banks; the length of curve II was about 1.4 km, and there were navigation obstacles at the concave bank on the right-hand side of the curve and the downstream riverbed. The navigable scale of the continuous curved channel in this study was 1.4 m × 35 m × 270 m (navigation depth × navigation width × bending radius). In the mathematical model test, the test sections upstream and downstream were provided with 16 m and 9 m transition sections to ensure that the water flow in the test section was constant and fully developed, as shown in Figure 1.

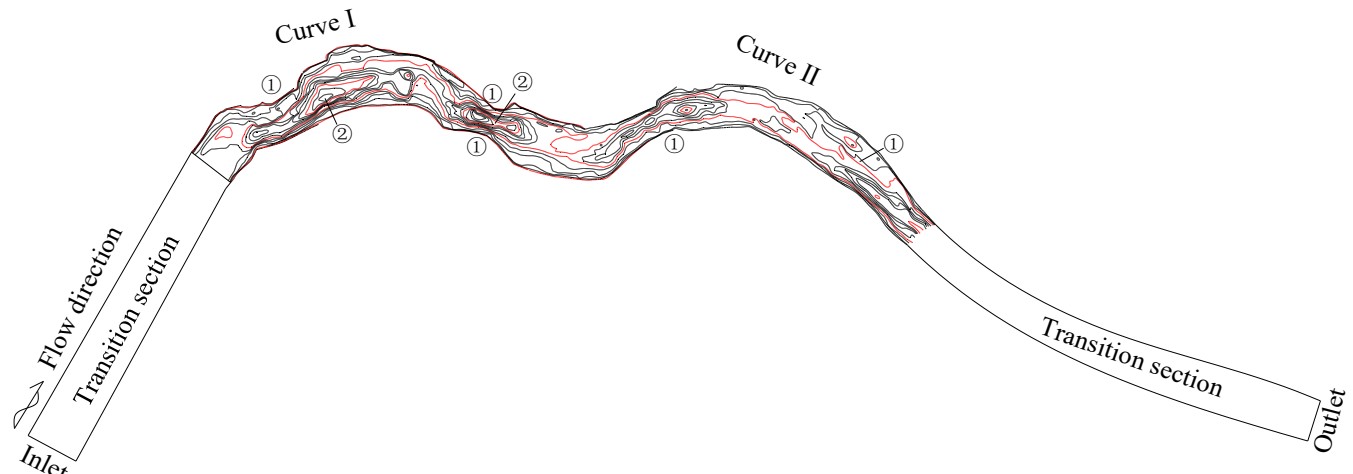

**Figure 1.** Diagram of the continuous curved river channel (① represents an obstruction point; ② represents low depression in the riverbed).

### 2.4. Experimental Conditions

In this study, representative flood flow (3410 m³/s), mid-water flow (1520 m³/s), and dry-water flow (374 m³/s) during the annual average flow were selected as the main parameters for the numerical simulation. The flood flow represents the flow corresponding to the highest navigable water level (1 in 3 years); the mid-water flow represents the flow corresponding to the 1.2-m-high remediation level; and the dry water flow represents the flow corresponding 95% of the guaranteed minimum navigable water level.

There were seven working conditions in this study: a prototype working condition for providing hydrological verification data, A1 (1520 m³/s), and six working conditions of mathematical model tests, i.e., natural conditions, L1 (374 m³/s), L2 (1520 m³/s), and L3 (3410 m³/s); and M1 (374 m³/s), M2 (1520 m³/s), and M3 (3410 m³/s) after river regulation. The working condition arrangement is shown in Table 1.

**Table 1.** Experimental parameters under different working conditions.

| Prototype Condition | | Mathematical Model Test Conditions | | | |
|---|---|---|---|---|---|
| Natural Conditions | | Natural Conditions | | After Remediation | |
| Working Condition | Flow (m³/s) | Working Condition | Flow (m³/s) | Working Condition | Flow (m³/s) |
| - | - | L1 | 374 | M1 | 374 |
| A1 | 1520 | L2 | 1520 | M2 | 1520 |
| - | - | L3 | 3410 | M3 | 3410 |

The hydrographic indicators of navigability are shown in Table 2. The hydrographic indicator of navigability means the theoretical maximum value of flow velocity suitable for ship navigation at different water level gradients.

**Table 2.** Hydrographic indicators of navigability.

| Gradient (‰) | 1.5 | 2 | 3 | 4 | 5 | 6 | 7 |
|---|---|---|---|---|---|---|---|
| Velocity (m/s) | 4 | 3.8 | 3.5 | 3.2 | 2.9 | 2.5 | 2.1 |

## 3. Results

### 3.1. Feasibility Analysis of the Mathematical Model

Model feasibility verification is an important part of model testing. The reasonableness of the model construction directly affects the accuracy of the study conclusions. A con-

tinuous curved river channel with a complex appearance and obstructions to navigation, such as obstruction points, leads to complex flow conditions. To ensure the accuracy and reliability of the results, the authors used hydrological data provided by a natural river prototype to verify the feasibility of the numerical model.

Mathematical model feasibility analyses are presented in Figures 2 and 3 and Tables 3 and 4. The results of the feasibility validation of the mathematical model show that the numerical simulation and the prototype water levels and flow velocities are in good agreement. The water levels were verified to be close to the value of 1. The maximum error value was 3%. There were only five locations where the flow velocities were in error with the natural flow velocities (error values greater than 10%), with a maximum error value of 13.8%, and in the rest of the locations, the flow velocities were in good agreement.

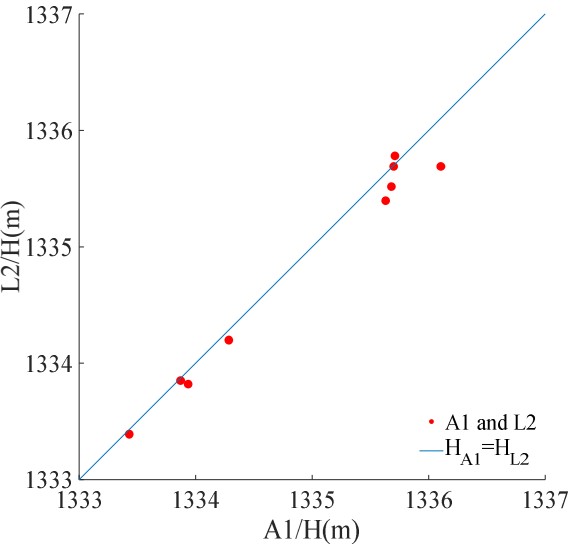

**Figure 2.** The water level fitting diagram.

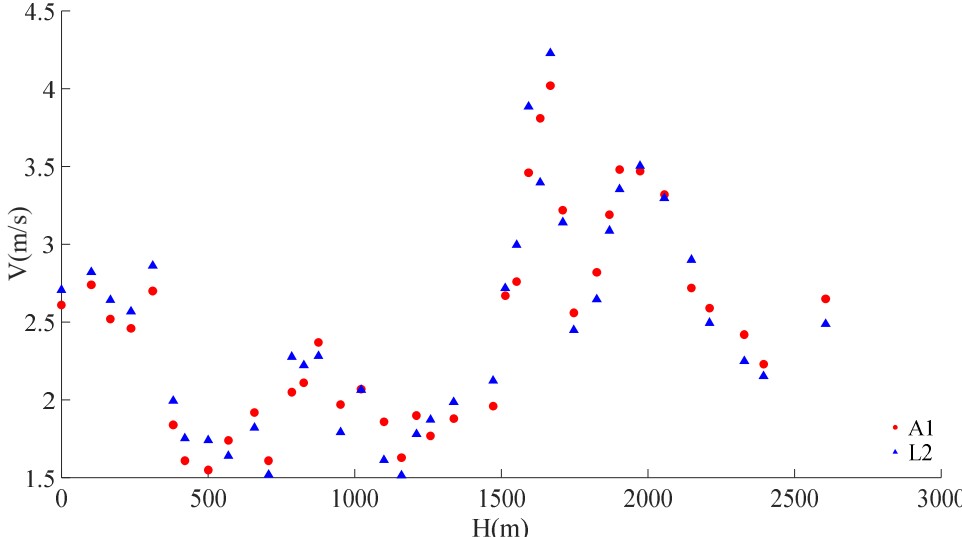

**Figure 3.** Flow velocity verification.

**Table 3.** Water level error analysis.

| Location | Error Value | Location | Error Value |
|---|---|---|---|
| C1 | −0.5% | C6 | 1.2% |
| C2 | 0.1% | C7 | 1.8% |
| C3 | 0.2% | C8 | 0.6% |
| C4 | 0.3% | C9 | 0.9% |
| C5 | 3.1% | | |

**Table 4.** Flow velocity error analysis.

| H (m) | Error Value | H (m) | Error Value | H (m) | Error Value |
|---|---|---|---|---|---|
| 0 | −3.73% | 876.6821 | 3.69% | 1667.3218 | −5.16% |
| 101.8836 | −2.96% | 952.0129 | 9.00% | 1709.6348 | 2.45% |
| 166.9831 | −4.83% | 1022.0596 | 0.28% | 1747.4577 | 4.35% |
| 237.3158 | −4.38% | 1100.1048 | 13.22% | 1825.6284 | 6.17% |
| 311.4188 | −6.02% | 1159.6542 | 7.08% | 1868.7288 | 3.21% |
| 381.3038 | −8.39% | 1210.8155 | 6.25% | 1903.3225 | 3.61% |
| 420.921 | −8.94% | 1258.7356 | −5.78% | 1973.2406 | −0.97% |
| 500.7897 | −12.32% | 1337.7123 | −5.63% | 2056.0031 | 0.72% |
| 569.4314 | 5.71% | 1472.2932 | −8.39% | 2148.2125 | −6.63% |
| 658.2227 | 5.11% | 1513.4778 | −1.80% | 2210.2138 | 3.66% |
| 706.4791 | 5.62% | 1552.1245 | −8.57% | 2328.5594 | 7.02% |
| 785.4862 | −11.08% | 1593.023 | −12.25% | 2394.4765 | 3.48% |
| 826.5563 | −5.36% | 1632.6447 | 10.85% | 2606.0386 | 6.09% |

*3.2. Numerical Simulation of Natural Rivers*

The smooth navigation of ships and the stability of fish survival environments were the focus of this study on the construction of ecological waterways. The authors, therefore, assessed the conditions of ship navigation and the fish survival environment by analyzing the hydraulic characteristics within a natural continuous and continuously curved river channel. The research on the hydraulic characteristics of natural river flow mainly included the velocity, water level, and flow pattern.

3.2.1. Flow Velocity

The distribution of channel velocity along the flow direction is shown in Figure 4. By observing the velocity change along the channel, it was concluded that the velocity in the channel fluctuated under different working conditions. The fluctuation range of flow velocity in curve II was significantly higher than in curve I. The reason is that the transition section was connected upstream of curve I, the smooth flow entered the bend, and the flow velocity changed gradually. Navigation obstruction points and low-lying riverbeds were situated in the bend; therefore, the flow velocity fluctuations in curve II became larger than in curve I after long-distance transportation. Under working conditions L2 and L3 upstream of bending curve II, the flow velocity fluctuated the most, the flow velocity increased rapidly, and there was maximum flow velocity at this position. The maximum flow velocities were 4.3 m/s and 4.8 m/s, respectively. This was because the flow was blocked by the bank slope at the continuous bend, consuming kinetic energy, and reducing the flow velocity. Therefore, the flow velocity at curve I was low. With the continuous inflow of upstream water, the flow at the continuous bend accumulated more potential energy. At a certain position of the continuous bend, the potential energy reached the threshold, the potential energy was converted into kinetic energy, and the flow velocity increased. The maximum velocity position of working condition L1 moved downstream of curve II, indicating that the maximum velocity fluctuation position moved downward with a decrease in the incoming flow.

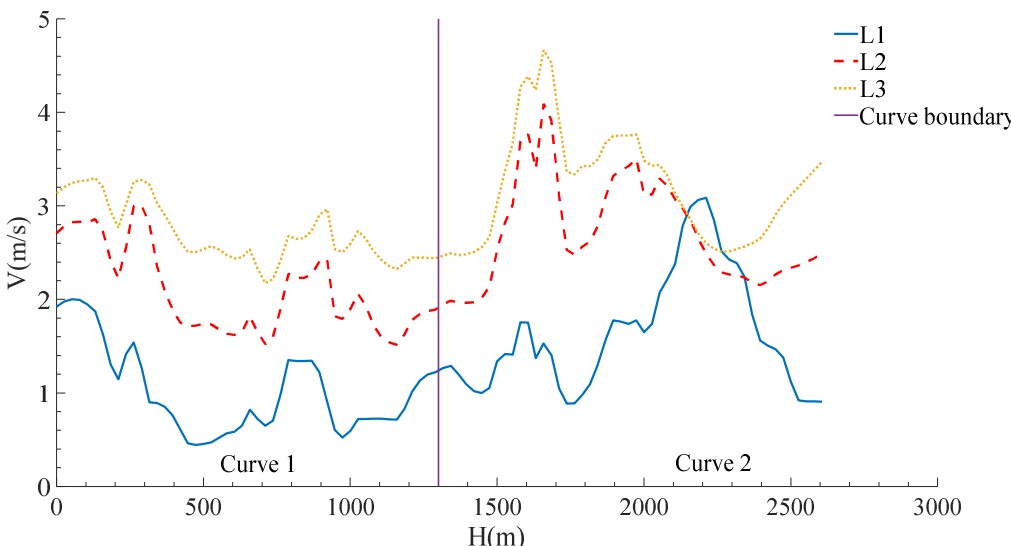

**Figure 4.** Flow velocity diagram of the river channel under natural conditions.

### 3.2.2. Water Level and Depth

The distribution of channel water levels along the flow direction is shown in Figure 5. By analyzing the water level and water level gradient along the channel, it was concluded that under different flow conditions, the water level change in the main channel of the river was relatively gentle, and the water level decreased along the channel. The water level in curve II changed at the continuous bending position, and the water level gradient was the largest. Under working conditions L2 and L3, the maximum water level gradient reached 3.0‰ and 4.0‰. The reason was that the upstream flow enters the curved section, and the obstruction of bank slopes on both sides of the river formed backwater, resulting in a reduction in the downstream flow speed, the conversion of kinetic energy into potential energy, and the short "detention" of the flow. When the "retention" position rose, the water level gradient increased, similar to the phenomenon of "hydraulic jump"; the maximum position of the water level gradient was the same as the maximum position of velocity fluctuation. Under working condition L1, the position with the maximum change in water surface position and the position with the maximum water level gradient moved downstream, indicating that the position with the maximum water level gradient moved downward with the decrease in the incoming flow.

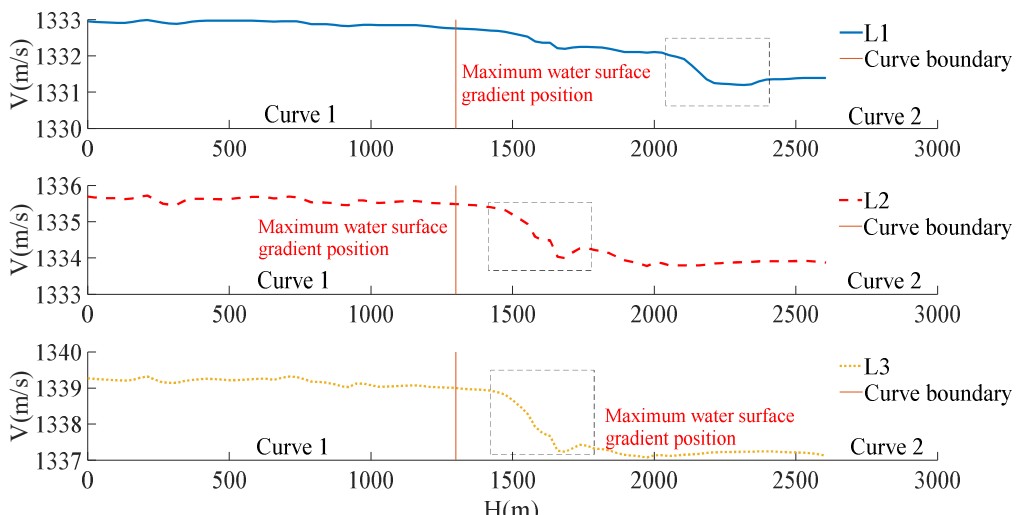

**Figure 5.** Water level diagram of river channels under natural conditions.

Flow velocity and water level gradient together affect the normal navigation of a ship. Under the same flow conditions, a large water surface drop had a negative impact on the navigation of the ship. Table 2 shows the theoretical maximum values of flow velocity suitable for ship navigation at different water level gradients. Referring to Table 2, when the water level gradient is 3.0‰ and 4.0‰, the maximum flow velocity suitable for ship navigation in the channel is 3.5 m/s and 3.2 m/s, respectively. In contrast, the maximum flow velocities in working conditions L2 and L3 (3.0‰ and 4.0‰) were 4.3 m/s and 4.8 m/s. Such conditions do not meet the requirements for ship navigation.

The water depth study mainly focused on the small flow condition of working condition L1. Under working condition L1, obstruction points on both banks and riverbed in curves I and II extended to the channel. Obstruction points lead to the water depth not meeting the navigable requirements for ships, as shown in Figure 6.

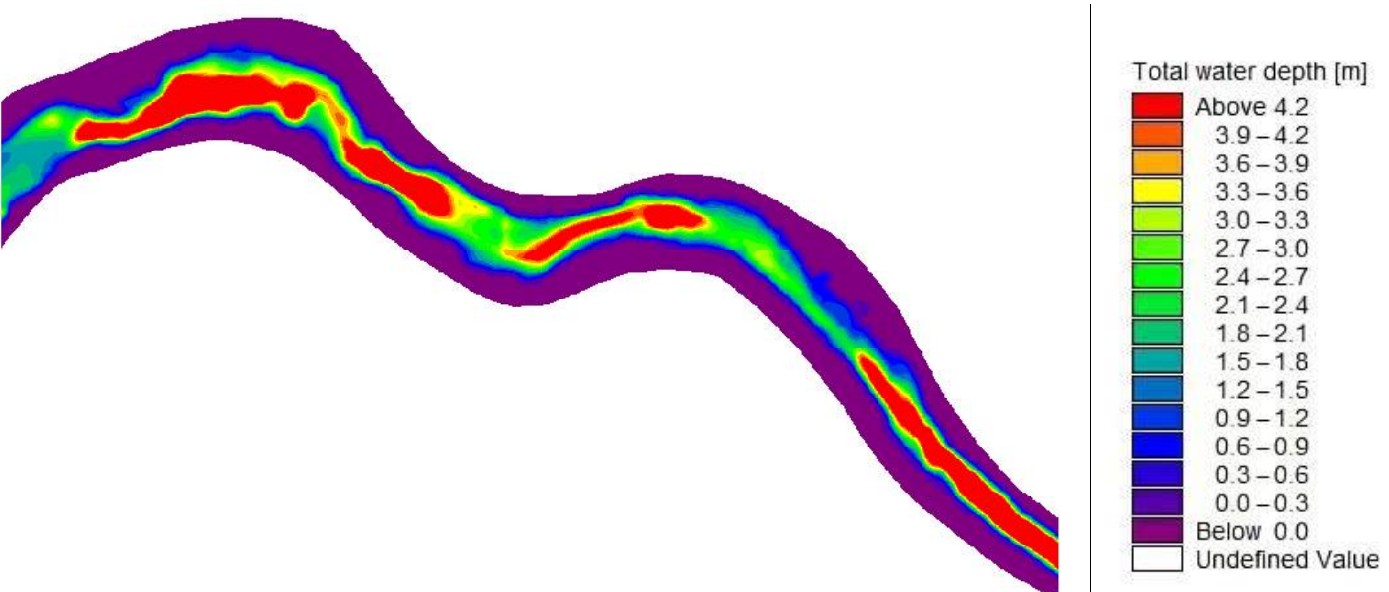

**Figure 6.** Water depth diagram of a natural river channel (working condition L1).

### 3.2.3. Flow Pattern

The flow pattern in the channel affects the safe navigation of ships, and poor flow patterns can easily lead to adverse effects such as ship yawing. Therefore, analysis of the channel flow pattern is also crucial.

This is shown in Figure 7. The continuous curved channel shows a turbulent flow pattern under different operating conditions. The reason for this is that in bend 1, the presence of obstructions on both banks causes the flow to be deflected, and as the flow increases, the influence of the obstructions on the flow diminishes, with a negative correlation between the angle of deflection and the flow. In curve II, the flow pattern was disturbed by the combined effect of the continuous bend and the obstruction. As the flow increased, the degree of turbulence was negatively correlated with the flow rate.

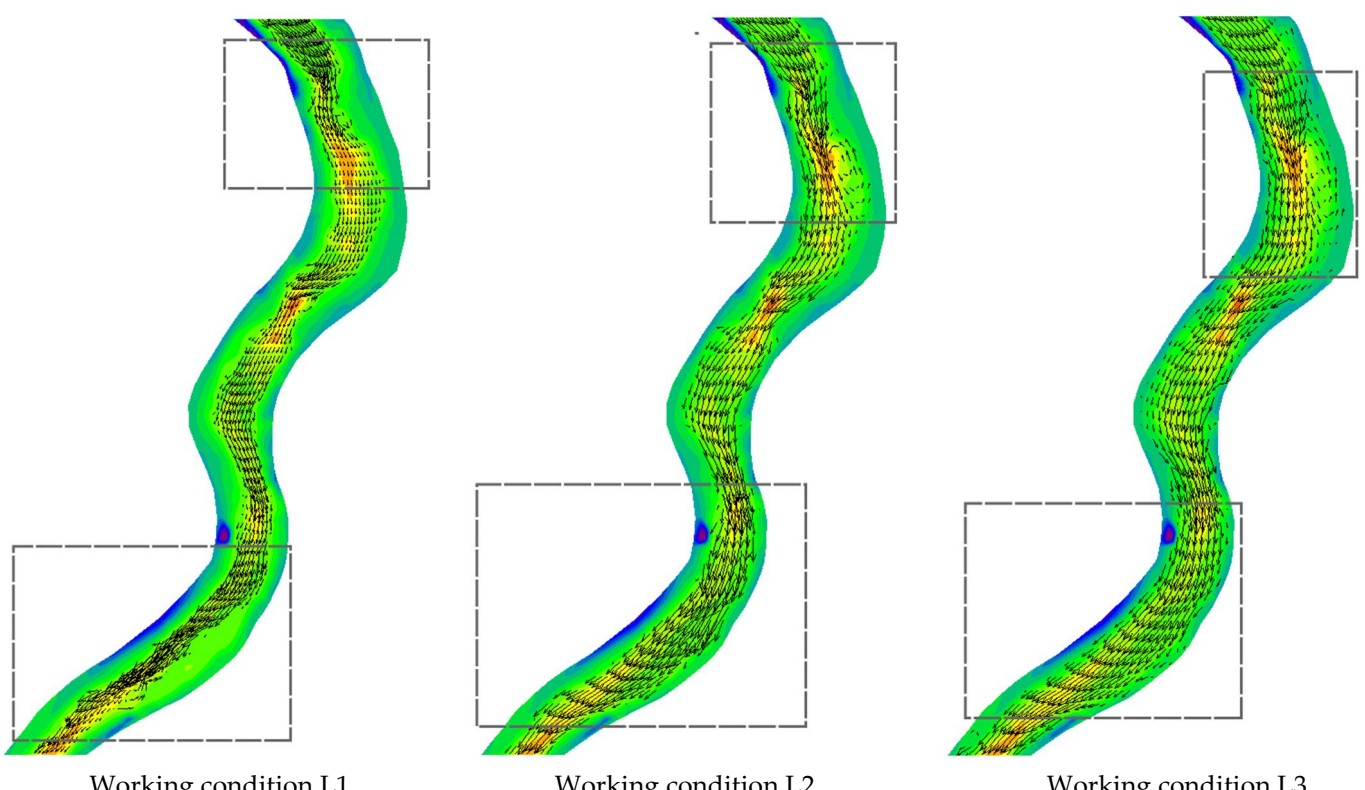

Working condition L1.　　　　　Working condition L2.　　　　　Working condition L3.

**Figure 7.** Water flow pattern diagram.

### 3.2.4. Fish Living Environment

The fish species analyzed in this study was grass carp, one of the "four major fish", which occupies a very important position in freshwater fisheries. It has a habit of growing in lakes, spawning in rivers, and migrating between rivers and lakes. It is a typical semi-migratory fish of the rivers and lakes. Migration is the result of long-term adaptations by fish to external environmental conditions. Fish migrate and change living waters to meet the needs of different living periods. Currently, the most commonly used indicators to reflect the swimming ability of fish are induced velocity, critical swimming speed, and burst swimming speed. The induced velocity is the minimum current speed at which fish can discern the direction of the current; this reflects the fish's ability to perceive the current. Typically, the induced flow velocity is taken to be low, at around 0.2 m/s. The critical swimming speed is also called the maximum sustainable swimming speed, which reflects the long-term swimming ability of fish. Explosive swimming speed is the maximum speed that fish can reach, which is of great significance for fish to avoid hazards, cross obstacles, and pass through high-speed and high-turbulent flow areas. In this study, the critical swimming speed was mainly used to analyze the living environment of grass carp.

Hu [26] found a correlation between the body length of grass carp and their critical swimming speed. The critical swimming velocity increased with the increase in body length and increased power function: $U_{crit} = 25.84L^{0.63}$, where $U_{crit}$ represents the critical swimming speed (unit: cm/s) and L represents the body length of grass carp (unit: cm). When the current velocity exceeded 60% of the critical swimming velocity of fish, the current conditions were not suitable for the long-term survival of fish, and the upstream migration rate of fish decreased. At the same time, water flow velocities below the induced flow velocity (v < 0.2 m/s) were not within the appropriate flow velocity range for grass carp backflow. The size of adult grass carp is generally between 30 and 80 cm, and the critical swimming speed was calculated to be about 2.2~4.1 m/s. The flow velocity corresponding to 60% of the critical swimming speed was about 1.3~2.5 m/s. Therefore, the flow velocity suitable for fish survival is 0.2 m/s < v < 2.5 m/s.

Figure 8 shows the river flow velocity distribution analysis under working conditions L2 and L3, where the velocity area with v > 2.5 m/s accounts for a large proportion, accounting for 45% and 25%, respectively. The river flow environment is not suitable for the long-term survival of grass carp.

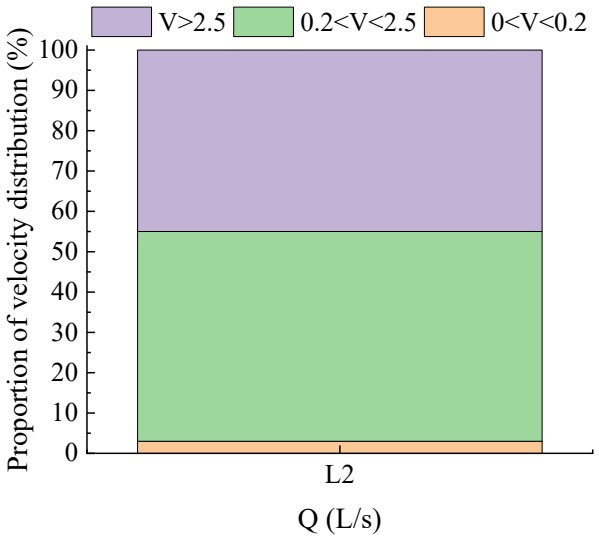 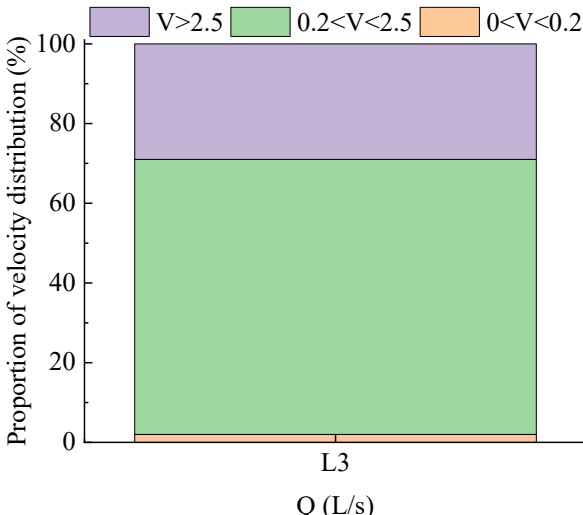

**Figure 8.** Proportion diagram of velocity distribution at all levels of the river channel under different flow conditions.

In summary, the hydraulic characteristics in the natural continuous curved channel were analyzed using vessel navigation and the fish survival environment as research elements. The following conclusions were drawn from the analysis of channel depth, flow velocity, flow regime, and water level gradient: under operating condition L1, the water depth in the channel did not meet the minimum requirements for vessel navigation. Under operating conditions L2 and L3, the water level gradient and flow velocity were high, and the flow regime was complex. These factors are not conducive to the safe navigation of vessels. Additionally, the high flow velocities at L2 and L3 and the flow velocity Shannon diversity index indicated that the river could not provide a long-term stable living environment for grass carp; therefore, the natural river was not conducive to the sustainable development of an ecological channel.

### 3.3. Numerical Simulation Analysis after River Regulation

The unstable flow characteristics of the continuously curving river make it impossible for ships to navigate safely, and do not provide a stable environment for fish to survive. The authors analyzed the factors hindering navigation, and propose a targeted channel regulation scheme. The dredging treatment scheme was adopted to remove the obstruction points; the riverbed elevation was low and concave, and the regulatory scheme mainly filled the groove. The main purpose of trench filling and dredging is to level the riverbed and bank slope and smooth the water flow. In order to reduce the water surface gradient along the way, submerged dams are built downstream of the continuous curved river to raise the water level. The remediation scheme is shown in Figure 9.

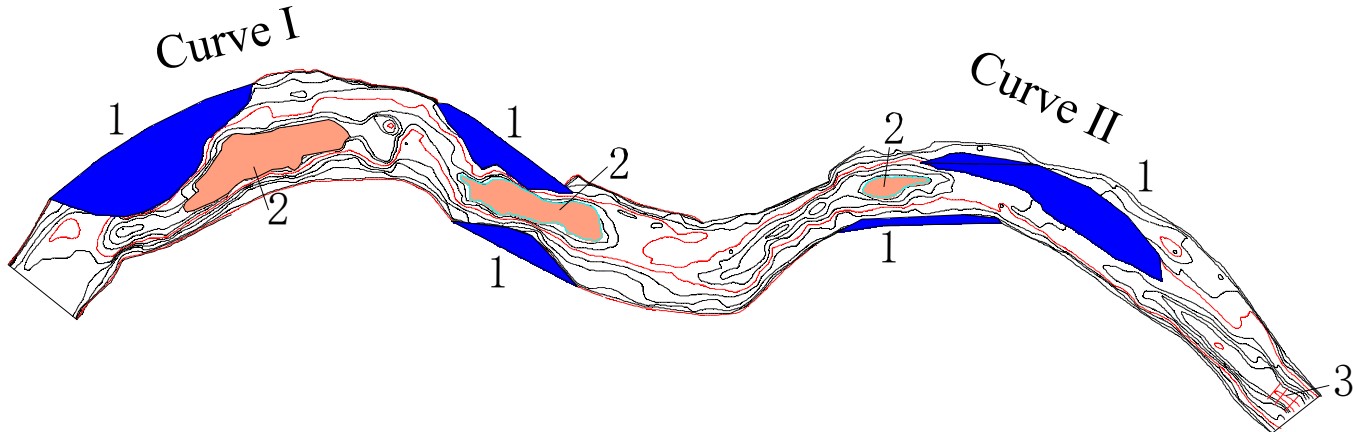

**Figure 9.** Layout of the treatment scheme (1 indicates dredging; 2 indicates a filling groove; 3 indicates the construction of a submerged dam).

### 3.3.1. Flow Velocity

The riverbed topography along the curved river channel had been greatly improved through river regulation, and the riverbed and bank slope were flat. By analyzing the flow velocity of the river after regulation, it can be concluded that the flow velocity fluctuation was significantly improved compared with that before regulation, and the flow velocity along the river was smooth, especially in the continuous curved river section. Under working condition M2, the maximum velocity decreased from about 4.3 m/s to 3.0 m/s; under working condition M3, the maximum velocity decreased from about 4.8 m/s to 3.5 m/s, and the maximum velocity decreased greatly. The decrease in the maximum velocity indicated that the riverbed leveling treatment and the removal of the river channel obstructing points reduced the obstructing effect of the curved river segment on the upstream flow and the scouring of the riverbed by the water flow. The potential energy threshold was lowered, speeding up the energy conversion between potential and kinetic energy. The improvement effect of the flow rate became increasingly obvious with the increase in flow rate, as shown in Figure 10.

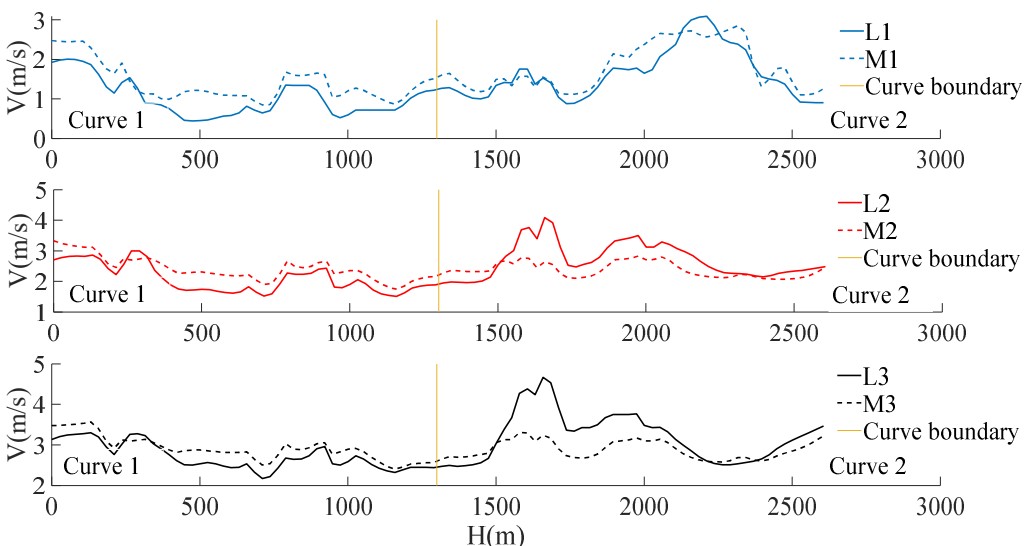

**Figure 10.** River velocity chart after regulation.

By analyzing the proportion of flow velocity distribution in the channel before and after the regulation under conditions L2, M3, L3, and M3, it can be concluded that after the regulation, the navigation obstacles that hindered the flow were removed, the flow velocity

distribution in the channel was significantly improved, and the flow velocity that affected the normal navigation of ships was completely eliminated; the 2.0~3.0 m/s flow velocity was expanded, and the proportion of flow velocity distribution reached about 60%. The Shannon diversity index was used to analyze the flow velocity diversity. Under condition M2, the flow velocity diversity index decreased from 1.52 to 0.94; under condition M3, the flow velocity diversity index decreased from 1.67 to 1.21. The diversity of the flow velocity decreased, and the flow velocity change in the channel was relatively smooth, as shown in Figure 11.

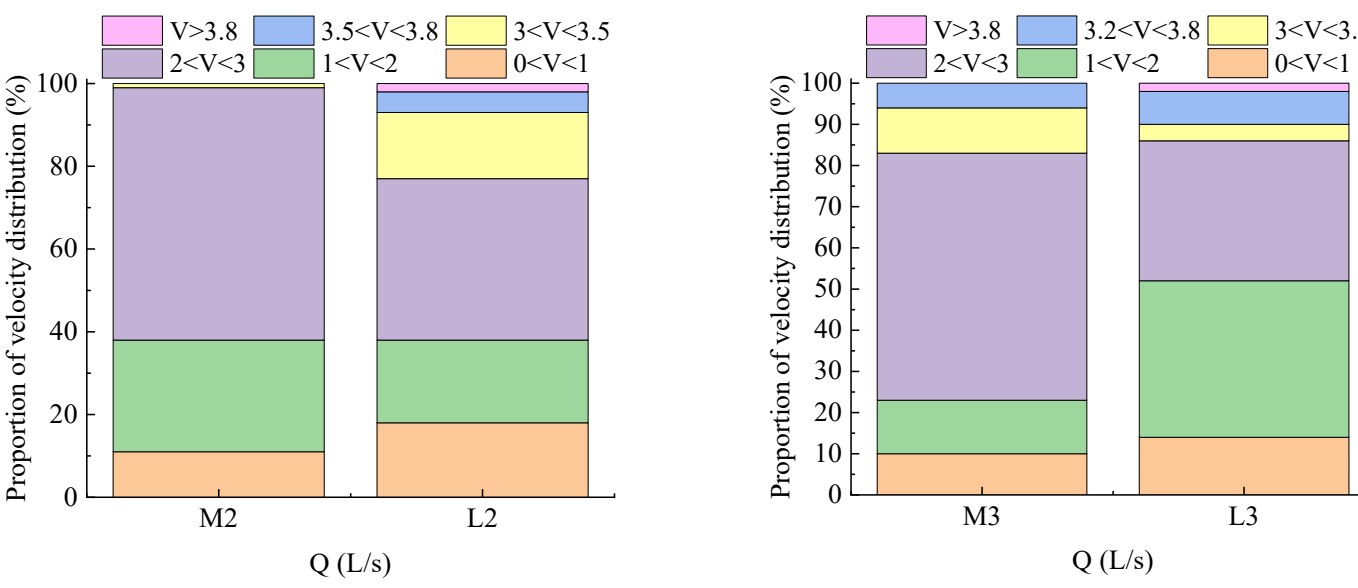

**Figure 11.** Proportion of flow velocity distribution in each layer of the river under different flow conditions.

### 3.3.2. Water Level and Depth

The water level analysis of the river after regulation shows that the water level changed greatly after regulation, and the maximum water level gradient decreased significantly. Under working condition M2, the maximum water level gradient decreased from 3.0‰ to 1.3‰, and under working condition M3, the maximum water level gradient decreased from 4.0‰ to 1.9‰. Reasons for water level gradient decreases include the removal of obstruction points in the curved river channel and the leveling treatment of the riverbed, which reduce the threshold of potential energy, accelerate the conversion between kinetic energy and potential energy of water flow at this position, reduce the water surface height, and stabilize the water level along the way. The improvement effect of the water surface height along the river was increasingly obvious with the increased flow. Additionally, at working condition M1, the water depth met the navigational requirements, as shown in Figures 12 and 13.

Referring to Table 2, when the water level gradient was 1.3‰ and 1.9‰, the maximum flow velocity in the channel suitable for ship navigation was not greater than 3.8 m/s. In M2 and M3 conditions, the maximum flow velocities in the continuous bend were 3.0 m/s and 3.5 m/s, respectively. These conditions met the requirements for ship navigation.

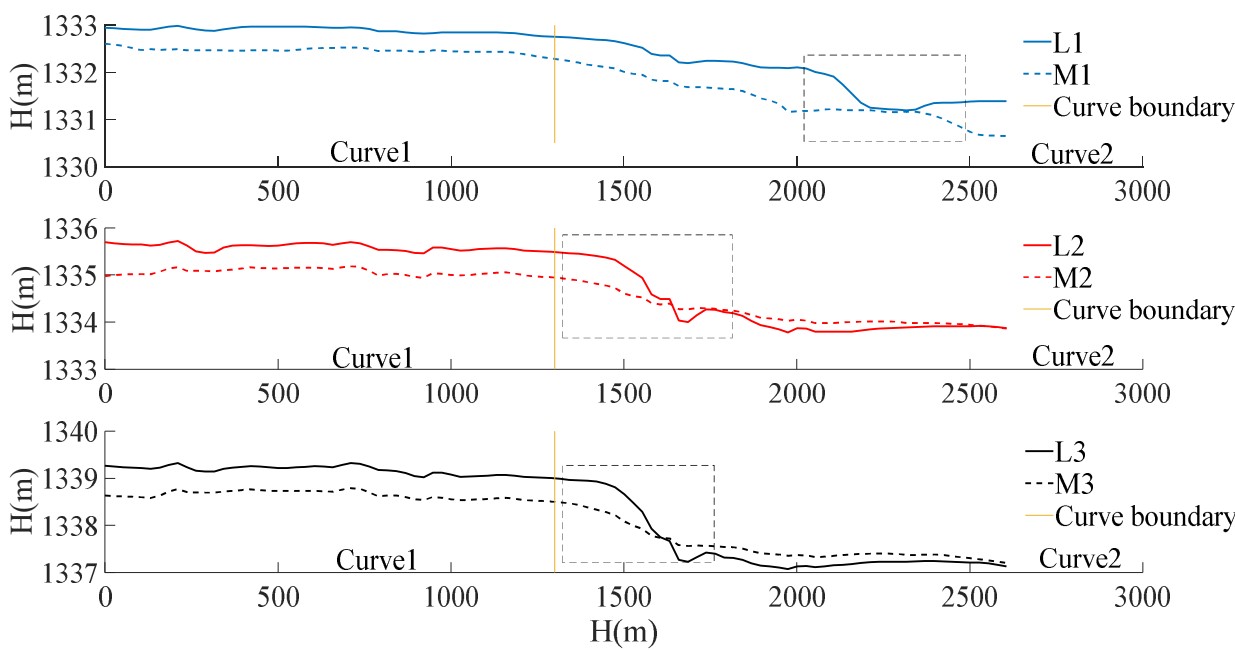

**Figure 12.** Water level of river channel after regulation.

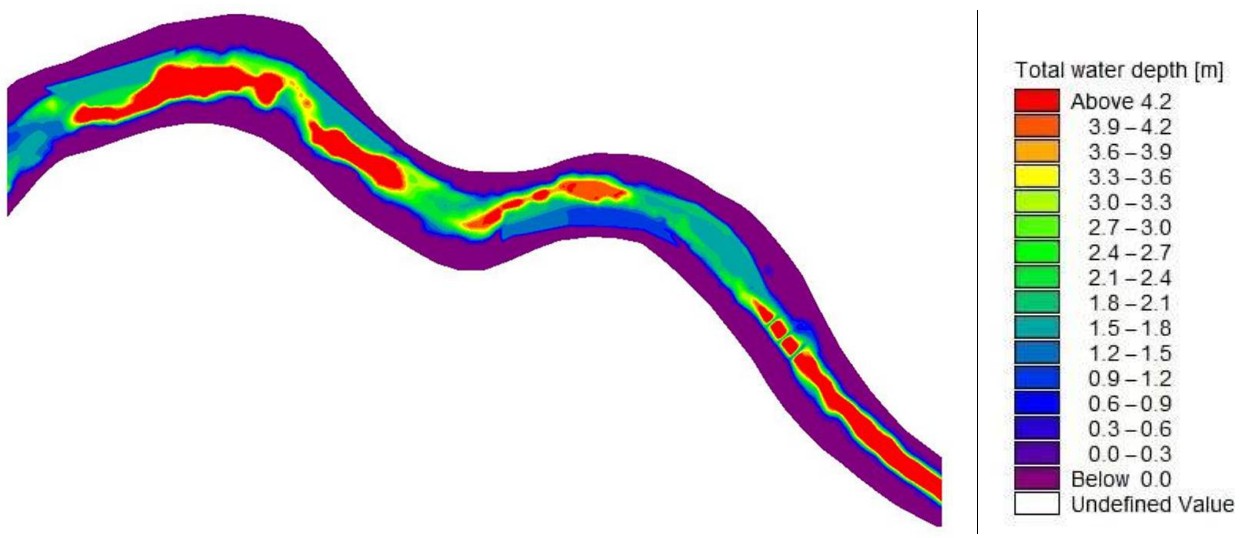

**Figure 13.** Water depth diagram of the river channel after regulation (working condition M1).

### 3.3.3. Flow Pattern

Through the analysis of the flow pattern of the river after regulation, it can be concluded that, in working conditions M1, M2, and M3, the removal of navigation obstruction points, the leveling of the riverbed, and other regulation means had improved the flow pattern of the continuous curved river reach. The large deflection angle of flow direction and disordered flow pattern were eliminated. The water flow was smooth and met the navigation requirements of ships, as shown in Figure 14.

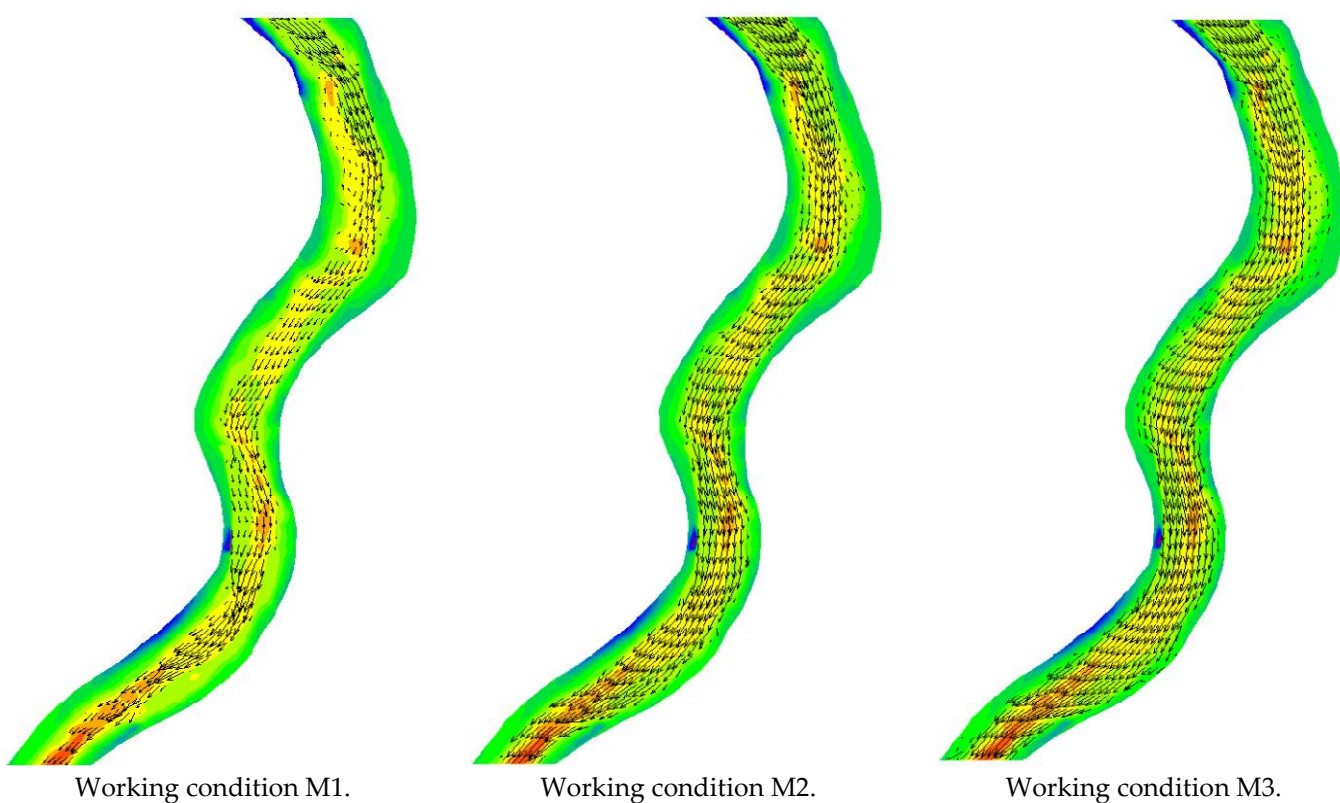

Working condition M1.  Working condition M2.  Working condition M3.

**Figure 14.** River flow pattern after regulation (working condition M1).

### 3.3.4. Fish Living Environment

By analyzing the fish living environment in the natural river after regulation, it can be concluded that, under working conditions M2 and M3, at curve II, the hydraulic characteristics, such as flow velocity, water surface gradient, and flow pattern, were improved after regulation. Under condition M2, the proportion of flow velocity conditions suitable for fish survival increased from 52% to 68%. The proportion did not change significantly under condition M3, at about 50%. This is because the river regulation eliminated large flows and smoothed the flow velocity. Continuous meandering channels provide a stable living environment for fish, and the living environment for fish here was largely improved, as shown in Figure 15.

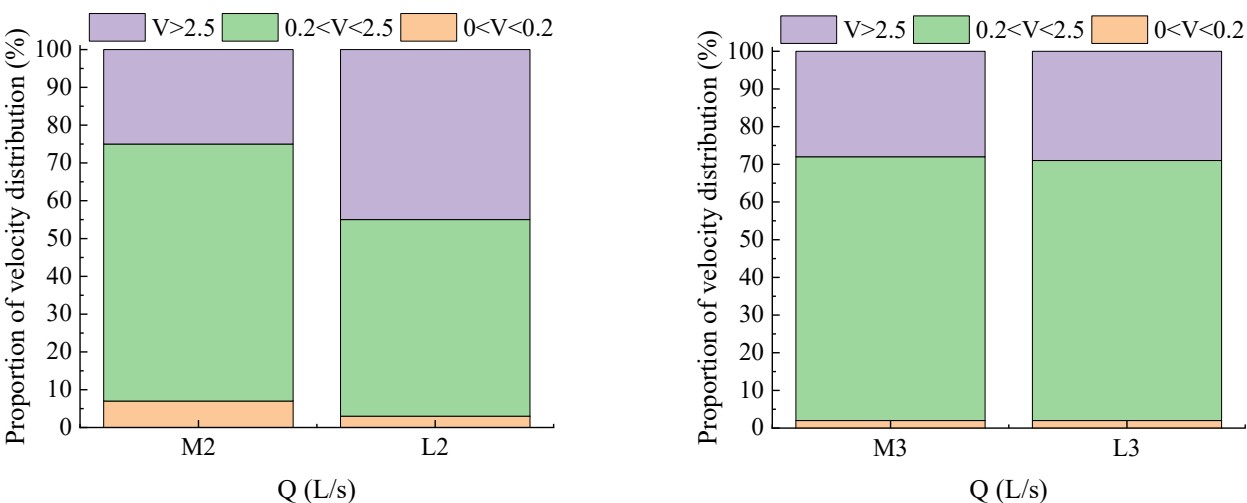

**Figure 15.** Proportional distribution of regulated flow velocities in each layer of the river.

In summary, the hydraulic characteristics of the remediated continuous curved river channel have been improved to meet the requirements of ship navigation and fish survival, and the construction of an ecological waterway has been effective. Through the numerical simulation of the continuous curved river channel, the authors found that among the continuous curved river channels, the curved river connection exhibited the most complex water characteristics, with sharp fluctuations in flow velocity and water level, which seriously affects the navigation of ships and the stable survival environment of fish, and is a key area for remediation in the construction of eco-channels. This study also provides basic research for the construction of ecological waterways in continuous curved rivers.

## 4. Conclusions

This study used a natural river channel as the research background, and numerical simulation (using the numerical simulation software, Mike21) as the research method to analyze the flow velocity, water-level-specific drop, water depth, and fish living environment before and after continuous curved river channel remediation, and the following conclusions have been drawn.

The hydraulic flow characteristics of the continuous bend section of a natural river channel are complex. At successive bends, the bank slope and the riverbed influence the hydraulic characteristics of the flow. The flow upstream of the bend is blocked in the bend section, the flow velocity decreases, the kinetic energy of movement is converted into potential energy, the water level gradually rises, and the potential energy increases, resulting in a large water level gradient at the bend. When the potential energy exceeds a threshold value, the potential energy begins to convert to kinetic energy and the flow velocity increases inward. This phenomenon becomes more and more pronounced as the inflow increases. As a result, under high-flow conditions, the flow velocity and water level gradient are too large, with maximum flow velocities of 4.3 m/s and 4.8 m/s at operating conditions L2 and L3, respectively, corresponding to water level gradients of 3.0‰ and 4.0‰, which is not conducive to vessel navigation and survival. Under L1 condition, the water depth was not adequate for vessel navigation due to the presence of obstruction points in the channel. The existence of obstruction points leads to a large angle of deflection in the water flow, and the flow pattern at the location of large flow velocity is complex and harsh, as is the water level gradient.

By analyzing and summing up the impediments to continuous bends, a targeted river management program is proposed, mainly by dredging, filling the channel, and constructing a submerged dam downstream. The main objective is to smooth the flow of water, reduce the slope of the water level in the continuous bend, and raise the water level. The hydraulic characteristics of the river channel are effectively improved. When the upstream flow enters the bend, the water-blocking effects of the banks and the riverbed are reduced, the potential energy threshold is lowered, the energy conversion between potential and kinetic energy is accelerated, and the water depth increases. In condition M1, the water depth in the channel meets the navigability requirements. The maximum flow velocity is 3.0 m/s and 3.5 m/s for working conditions M2 and M3, respectively, corresponding to a water level gradient of 1.3‰ and 1.9‰, whereas the river flow Shannon diversity index decreases from 1.52 and 1.67 to 0.94 and 1.21, respectively, and the flow regime is improved. The modified hydraulic flow characteristics met the vessel's navigational requirements.

Grass carp was the representative fish in this study. Analysis of the fish habitat before and after the continuous bend channel rehabilitation using the Shannon diversity index concluded that in the natural channel, with increasing flow, the flow velocity at the continuous bend was too high and the area of the channel unsuitable for long-term fish survival was too large to provide a stable habitat for fish. After the remediation, improvements in flow velocity, water surface gradient, and flow regime provided a stable habitat for fish. In working condition M2, the proportion of flow conditions suitable for fish survival increased from 52% to 68%, and the range of fish activity also increased. The regulating effect was obvious.



In continuous curved river channels, the curved river connections have the most complex water characteristics, with sharp fluctuations in flow velocity and water level, which seriously affect the navigation of ships and the stable survival environment of fish, and are, thus, key areas for remediation in the construction of eco-channels. This study also provides basic research for the construction of ecological waterways in continuous curved rivers.

However, despite these important findings, there are some limitations to this study. The authors did not analyze the sediment movement of the river. In addition, the authors only carried out two-dimensional rather than three-dimensional numerical simulations. In the fish habitat study, the authors only considered the effect of velocity and did not consider the effect of other factors on fish habitats.

In the future, the authors will carry out experimental studies of three-dimensional water-sediment numerical simulations of continuously curving rivers, as well as more in-depth studies of the factors influencing the fish habitat.

**Author Contributions:** Software, J.L. and P.W.; validation, J.L. and P.W.; formal analysis, J.L. and P.W.; resources, J.L. and P.W.; data curation, J.L. and P.W.; writing—original draft preparation, J.L. and P.W.; writing—review and editing, J.L. and P.W.; funding acquisition, P.W.; investigation, J.L., P.W., M.W., J.H., and F.Z. All authors have read and agreed to the published version of the manuscript.

**Funding:** This research was funded by the Science and Technology Research Program of Chongqing Municipal Education Commission (Geomorphic changes of the upper reaches of the Yangtze River under new water and sediment conditions and their impacts on habitat conditions), grant number: KJQN-201900745.

**Institutional Review Board Statement:** Not applicable.

**Informed Consent Statement:** Not applicable.

**Data Availability Statement:** Not applicable.

**Conflicts of Interest:** The authors declare no conflict of interest.

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
