# Peer review of "Numerical Simulation of the Hydraulic Characteristics and Fish Habitat of a Natural Continuous Meandering River"

_sustainability, doi:10.3390/su14169798_

Round 1

Reviewer 1 Report

Dear Authors,

the aspects listed below need to be extensively elaborated to rise the level of your article to the Journal standards.

1) The abstract should be more concise and better structured. It has to provide a brief introduction of the background and then it has to explain the literature gap covered, the methodologies applied and the results achieved.

2) The state of the art is actually really concise and not focused on the scope of the paper. An accurate evaluation of the references strictly referred to the methodology and results presented should be provided. As a consequence, this would clearly define the lacks of the current knowledge. In the present form an original contribution of the research does not emerge from reading the text.

3) The effects of the assumptions of the proposed methodology on the outcomes need to be discussed in Chapter 2. Including a dedicated subsection after the one numbered 2.2 is suggested.

4) The quality of figures and tables is not satisfactory. The labels are too tight compared to the size of the graphs and they are hard to read. Moreover, many of them have very low resolution. In addition, the Guidelines for the Authors should be followed to be compliant with the Journal requirements and make the formatting homogeneous.

5) The explanation of the results presented requires the introduction of their physical motivations. In the present form, the discussion is limited to a poor description of the data.

6) The number of figures should be limited to the ones depicting the most important outcomes. Actually, too many images are presented without detailing the reasons determining the different behaviours of the various configurations and working conditions investigated.

7) The Conclusions section needs to be focused on the key aspects of the model and the achieved results. Furthermore, all the statements should be supported by numerical data.

9) The quality of English must be improved. There are many mistakes and inconsistencies in the text.

10) The highlights of the paper should be included.

Author Response

请参阅附件。

Reviewer 2 Report

Manuscript ID: sustainability-1831562

Title: Numerical simulation of the hydraulic characteristics and fish habitat of a natural continuous meandering river

Remarks:

In this paper, the hydraulic characteristics of flow and fish survival environment are analyzed in a curved river channel. Then, the effect of a river management plan is simulated, and the hydraulic characteristics and fish's survival environment conditions were compared with the natural river channel features. Major changes are required prior the publication of this manuscript:

Abstract must be rewritten.

The objective and contribution of this work is not well established.

In the materials and methods section, hydrodynamics equations are mentioned (Equations 1 to 9). However, these equations are solved using simulation software Mike21. Therefore, these equations could be resumed or eliminated since no contribution is given.

Authors must describe the assumptions, considerations and the values of the coefficients used for numerical simulations. These values must be provided for the reproducibility and repeatability of the simulations.

Another critical input need for numerical simulations is the riverbed topography. How was this bathymetry obtained? An in-depth description about this bathymetry is needed.

Three working conditions for mathematical modeling were used: flood flow rate (3140 m3/s), mid-water flow rate (1520 m3/s), and dry water flow rate (374 m3/s) of the annual average flow. The procedures or criterion for calculating or determining these flow rates is not given.

Fitting diagrams are not enough for the verification of the feasibility of the mathematical model. Statistics indicators (measures) are needed.

I suggest including a table that shows the model coefficient values (eddy viscosity, the resistance coefficient, among others) used in this study. These values should be compared to coefficient values used in other simulation studies.

The description of the distribution of channel water levels under different flow conditions (hydraulic characteristic zonings) is well known in literature. However, no comparisons with other rivers are given.

Most of the results are described as a technical report instead of a scientific paper.

Too many figures are given.

English revision is needed.

Round 2

Reviewer 1 Report

Dear Authors,

all the points of criticism were adequately addressed during the review process.

Reviewer 2 Report

Accept in present form